# Can We Discover Truffle’s True Identity?

**DOI:** 10.3390/molecules25092217

**Published:** 2020-05-08

**Authors:** Staša Hamzić Gregorčič, Lidija Strojnik, Doris Potočnik, Katarina Vogel-Mikuš, Marta Jagodic, Federica Camin, Tea Zuliani, Nives Ogrinc

**Affiliations:** 1Department of Environmental Sciences, Jožef Stefan Institute, Jamova 39, 1000 Ljubljana, Slovenia; stasa.gregorcic@ijs.si (S.H.G.); lidija.strojnik@ijs.si (L.S.); doris.potocnik@ijs.si (D.P.); marta.jagodic@ijs.si (M.J.); tea.zuliani@ijs.si (T.Z.); 2Jožef Stefan International Postgraduate School, Jamova 39, 1000 Ljubljana, Slovenia; 3Department of Biology, Biotechnical Faculty, University of Ljubljana, Jamnikarjeva 101, 1000 Ljubljana, Slovenia; Katarina.VogelMikus@bf.uni-lj.si; 4Department of Food Quality and Nutrition, Research and Innovation Centre, Fondazione Edmund Mach, 38010 San Michele all’Adige, Italy; federica.camin@unitn.it; 5Center Agriculture Food Environment (C3A), University of Trento, via Mach 1, 38010 San Michele all’Adige (TN), Italy

**Keywords:** *Tuber*, species, stable isotopes, elemental composition, multivariate discriminant analysis, geographical origin

## Abstract

This study used elemental and stable isotope composition to characterize Slovenian truffles and used multi-variate statistical analysis to classify truffles according to species and geographical origin. Despite the fact that the Slovenian truffles shared some similar characteristics with the samples originating from other countries, differences in the element concentrations suggest that respective truffle species may respond selectively to nutrients from a certain soil type under environmental and soil conditions. Cross-validation resulted in a 77% correct classification rate for determining the geographical origin and a 74% correct classification rate to discriminate between species. The critical parameters for geographical origin discriminations were Sr, Ba, V, Pb, Ni, Cr, Ba/Ca and Sr/Ca ratios, while from stable isotopes *δ*^18^O and *δ*^13^C values are the most important. The key variables that distinguish *T. magnatum* from other species are the levels of V and Zn and *δ*^15^N values. *Tuber aestivum* can be separated based on the levels of Ni, Cr, Mn, Mg, As, and Cu. This preliminary study indicates the possibility to differentiate truffles according to their variety and geographical origin and suggests widening the scope to include stable strontium isotopes.

## 1. Introduction

Truffles (*Tuber* spp.) belong to the ectomycorrhizal fungi (EMF) that undergo a complex life cycle in association with various forest species. They are among the most prized ingredients in the culinary world and can fetch hundreds to thousands of Euros per kilogram, depending upon the species and size [1]. Europe accounts for 85% of the world export market, where the most sought-after black and white truffles grow in France, Italy, Croatia, Slovenia, and Hungary. Among the different species of truffle, only three are commercially important: the white truffle (*Tuber magnatum* Pico), the black truffle (*Tuber melanosporum* Vittad.), and the summer truffle (*Tuber aestivum*). *Tuber magnatum* is the most valuable species, but its spread is limited to the limestone-rich floodlands of Italy and the Balkan peninsula, whereas *T. aestivum* is the most widely spread truffle species in Europe [2]. In Slovenia, truffles are located in areas of high ecological value and biodiversity due its geographic location combined with diverse relief structure, complex geology, substantial water resources and modified Mediterranean, continental and mountainous climates [3]. A combination of these parameters contributes to a considerably rich mycodiversity in the country and experts estimate that there are 16 species of truffle in Slovenian forests. These include the white truffles *T. magnatum*, *T. borchii* and *T. asa*, and the black truffles *T. aestivum* Vitt., *T. ubicatum*, *T. melanosporum* [4].

High truffle prices have led to several forms of adulteration. For example, 15 per cent of French truffles tested in 2012 were the more common, cheaper truffle species originating mainly from China. *Tuber borchii* can be visually confused with *T. magnatum* and sold as the latter. Another fraudulent practice involves the use of unripe fruiting bodies (ascocarps) of cheaper species in processed foods [5]. The extent of the fraud means that there is an urgent need to protect the truffle market and establish clear information on the truffle products’ link to its country of origin. Despite a large number of published studies on the ecology, genetics and cultivation of truffles [6,7,8], few studies have attempted to validate truffle authenticity. Of these, most used molecular approaches [9,10,11] and the analysis of volatile organic compounds in order to determine the authenticity and adulteration of truffles and truffle containing products [12,13,14,15]. 

Although stable isotopes and elemental fingerprinting has become increasingly important in establishing the authenticity and geographical origin of food products [16], to date, it has not been applied to truffles [17]. Studies where the stable isotope techniques have been applied were orientated towards investigating carbon isotope fractionation during sucrose decomposition [18], the ecophysiological relationship between truffles, soil and host plants [19], or assessing the mycorrhizal versus the saprophytic status of fungi using the natural abundance of carbon and nitrogen stable isotopes [20]. Habitats with different nutrient inputs and plant communities can show significant differences in overall carbon (^13^C/^12^C) and nitrogen (^15^N/^14^N) isotope ratios [21,22,23,24]. Geographical differentiation involving stable isotope composition have been performed for other fungal species. Based on the ^13^C/^12^C, ^15^N/^14^N, ^18^O/^16^O, and ^34^S/^32^S ratios in mushroom (*Agaricus bisporus*), it was possible to differentiate between specimens from six regions in Korea [25], while Puscas et al. [26] were able to distinguish samples from different regions of Transylvania using carbon isotope ratios of bulk fungi (^13^C/^12^C) and the hydrogen and oxygen isotope ratios (^2^H/^1^H, ^18^O/^16^O) in water extracted from the samples. It was also possible to obtain supplementary information about the geographical origin of truffles from the ^87^Sr/^86^Sr ratios, since almost the entire life cycle of the ascocarps takes place underground in the presence of the host tree. Importantly, the ^87^Sr/^86^Sr isotopic fingerprint remains unaltered up to the end product, even after processing, and hence provides an unique and well-established geographical tracer for several types of plant food product, such as rice [27], vegetables [28,29,30,31], cereals and mushrooms [32,33]. 

To date, only two studies have looked at the elemental composition of *T. magnatum* in order to assess potential differences in their assimilation and accumulation abilities [34,35]. *Tuber magnatum* appears to be the more competitive of the different varieties, being able to more efficiently assimilate/accumulate Cu, K, Na, P, and Zn. At the same time, *T. brumale* was more successful in accumulating/assimilating sulphur. Segneanu et al. [35] investigated antioxidant activity, total organic carbon as well as the levels of As, Cu, Pb, Zn, Mn, Fe and Ni in *T. magnatum* and *T. melanosporum*. Their results also show that *T. melanosporum* contains a high amount of C and Fe than *T. magnatum* Pico, while there was no difference in the levels of the other elements between two truffle species. 

The accurate discrimination of the geographical origin of truffles remains a critical issue because of unknown influence of genetic and environmental variations that affect their elemental and stable isotope composition. To overcome this lack of information, we performed a study to (i) characterise Slovenian truffle species for elemental and stable isotope composition; and (ii) explore the possibility of differentiating truffles according to species and geographical origin using multivariate statistical analysis. This study is part of the REALMed project (https://realmedproject.weebly.com/).

## 2. Results and Discussion

The present study gives insight into the elemental and isotopic composition of commonly cultivated *Tuber* species in Europe, while Chinese samples were collected from the local market. In this study, 58 samples of truffles from Slovenia (n = 33; *T. aestivum*, *T. brumale*, *T. magnatum*), Italy (n = 6; *T. aestivum*, *T. magnatum*, *T. melanosporum*), Croatia (n = 3; *T. aestivum*, *T. brumale*, *T. macrosporum*), Poland (n = 3; *T. aestivum*), Bosnia and Herzegovina (n = 2; *T. aestivum*), Spain (n = 3; *T. melanosporum*), North Macedonia (n = 5; *T. aestivum*, *T. mesentericum*), and China (n = 3; *T. indicum*) were considered. 

A summary of geo-environmental, climatic and host tree species information for *Tuber* species is presented in Appendix A. The elemental profiles and isotopic composition of light elements of truffles are presented in Appendix A, respectively (Appendix A). The most commonly evaluated elements in fungi were analysed, including Ca, Cd, Cu, Fe, Hg, K, P and Pb. Also. Al, As, Ba, Co, Cr, Cs, Mg, Mn, Na, Ni, Rb, S, Sr, V, and Zn, which are rarely evaluated, were determined. The data were then used to create a complete overview of elements for the average, minimum and maximum ranges of elements in different truffle species of wide geographical origin (Table 1). Furthermore, in total 33 different variables were included in multivariate statistical analysis to differentiate truffles according to the species and geographical origin. 

### 2.1. Elemental Composition

The mean concentrations of each element varies from species to species, which appears to be related to their geographical origin and growing conditions, such as soil characteristics, water availability and climate [36]. The results are presented in Table 1.

Thus, elemental composition can serve as a fingerprint for truffles, providing a useful marker for geographical classification. For this reason, samples were sorted into different groups based on species affiliation and geographical origin. 

Furthermore, the mobilization and redistribution of elements in truffle tissues are also important and will have a large impact on their heterogeneity [37]. The distribution of elements in truffles (*T. aestivum*) was recorded on the BL6b beamline, SLRI (Synchrotron light research institute, Thailand, Project proposal 3457). The results show that the microelements (Mn, Fe, Cu and Zn) and Ca accumulate mainly in the melanised layer of the truffle surface, while K accumulates in the core (Figure 1). Thus, the interpretation of our results is oriented mainly in the peridial layer of the fruiting bodies.

Taking into account that forests are complex systems, where large fluxes of essential and trace elements are balanced over time [36], we studied the possible differences or similarities in elemental composition relating to the geographical origin of samples of *T. aestivum* from Slovenia and abroad. The mean contents of elements determined in the peridial layer of investigated species decreased in the following order: K > P > Ca > S > Mg > Al > Fe > Zn > Na > Cu > Mn > Rb > Ba > Cd > Sr > Cr > V > Ni > Pb > As > Co > Cs > Hg. 

After applying an ANOVA test, significant differences (*p* < 0.05) in the levels of Na, Mg, S, Cu, and Ba between one or more countries were observed (Figure 2). The Croatian truffle samples differ from those of other countries by containing higher levels of Mg and Ba, and lower amounts of Na and Cu. High S and Sr contents are characteristic of the samples from Poland. 

In Slovenia, six main regions were identified: Sežana, Bloke, Rajndol, Spodnje Blato, Žlebič and Slovenian Istria. Truffles from Sežana exhibit higher concentrations of Al, Ca, V, Cr, Mn, Fe, Co, Ni, Pb, As and Sr, while concentrations of Mg, Rb, Cd and Hg were the highest at Bloke. On the other hand, the lowest concentrations of element were mainly observed at Rajndol and Žlebič. In Slovenian Istria higher concentrations of Na, K and Zn were observed, while concentrations of Ca and Cd are lower compared to other locations. The main reason for this difference is probably geological background and soil depth. For example, Sežana is located on calcaric flysch with eutric brown soil, while Žlebič and Rajndol are located on marine carbonate and clastic rocks. 

The content of individual elements in certain ectomycorrhizal fungal species is known to vary [38,39,40], while element contents in the fruiting bodies are species-dependent, which is consistent with findings in this study. For example, Ambrosio et al. [41] showed that the concentration of Cu, Zn, Sr, and Sb in Porchini mushrooms is very similar to that measured in soil layers, especially at the surface, while certain elements such as Cr and Ni had different distributions. Soil moisture may also have a significant effect on nutrient uptake, since the water phase is enriched in the more soluble Ca, Sr, Mg, Na, and K ions, whereas Al, Fe, and Ba, being less soluble, are enriched in the soil compartment. In addition, the elemental composition in truffles depend also on an accumulation and and assimilation capacity of the respective truffle species, which is also controlled by the host plant demand.

The ANOVA test also revealed significant differences in the amounts of Na, Mg, Al, P, S, Cr, Zn, and Ba in different species (Figure 3). The amounts of Na, Mg, Zn, Ba and S are significantly higher in *T. magnatum*, while the concentrations of P and S are significantly lower in *T. aestivum*. 

The Zn content of *T. magnatum* is twice that of the other truffle species, which is consistent with the results obtained for *T. magnatum* and *T. brumale* from Serbia [34]. In their study, *T. magnatum* appeared to be more competitive, being able to more efficiently assimilate/accumulate Cu, K, Na, P, and Zn. At the same time, *T. brumale* was more successful in assimilating/accumulating S. In the case of Cr, a significant difference was only observed between *T. melanosporum* and *T. magnatum*, with lower amounts in *T. melanosporum*. Except for Mg, P, Hg, and Sr, *T. indicum* had overall lower contents of elements. 

There have been numerous studies over past twenty years, particularly in Europe, examining the presence of heavy metals in ectomycorrhizal fungi and the results show heterogeneous behavior between species [42,43,44,45]. Ectomycorrhizal fungi tend to accumulate toxic elements such as Cd, As, Pb, and Hg, which may originate from both natural and anthropogenic sources [46,47]. The toxic effect of these elements seems to affect enzymes. The determination of the concentration of toxic elements in the truffle ascocarps is also essential for dietary intake studies. The range of toxic elements (As, Cd, Cr, Hg, Ni, and Pb) present in truffle samples tested in this study was lower than the permissible range for fungi (<0.5–5 mg/kg), indicating that their consumption is safe provided that it is occasional [48]. 

### 2.2. Stable Isotope Ratios of Light Elements

Appendix A gives the data for the stable isotope ratios of light elements. In this study, a wide range of *δ*^15^N values (1.8‰ to 19.6‰) are observed, while *δ*^13^C values were from −28.5‰ to −23.8‰. *Tuber magnatum* and *T. melanosporum* had the highest *δ*^15^N and *δ*^13^C values, respectively, while *T. aestivum* had the lowest. Figure 4 shows the relationship between *δ*^15^N and *δ*^13^C. These data are comparable with the literature data for EMF fungi [24,49,50]. A statistically significant difference in the *δ*^15^N values is found between different species and geographical location, while *δ*^13^C values showed no significant difference. The isotopic index Δ_CN_ = *δ*^13^C − *δ*^15^N, which allows the assignment of a mycorrhizal or a saprotrophic strategy for sporophore differentiation, ranged from −47.6‰ to −29.0‰, and suggests that *Tuber* does not exhibit a saprotrophic strategy. The limit between saprotrophic and symbiotic strategies is 24‰ [21]. This observation agrees with the findings of previous studies [49,50,51], but contradicts what the authors stated in handbooks related to truffle cultivation [52]. 

Although the molecular mechanisms for nutrient exchange between mycorrhizal symbionts are not clearly understood, the occurrence of mycorrhizal fungi is primarily controlled by the accessibility of primary nutrients; plants provide carbon, while nitrogen is derived from the soil, since ectomycorrhizas can uptake, reduce and metabolise nitrate [53]. The large variation in *δ*^15^N values, therefore, cannot be explained by taxonomic variation but rather by the nutrient source (*δ*^15^N_inorganic_ < *δ*^15^N_organic_) and soil depth of N acquisition (*δ*^15^N_shallow_ < *δ*^15^N_deep_) [22]. Also, the relative dependence of organic N pools appear to vary according to the location with latitude and altitude of mycorrhizal origin [54]. Another possible explanation for higher δ^15^N values is isotopic fractionation, which can occur during the transport of nitrogenous compounds from ectomycorrhizal fungi to their host plants [49]. 

As already mentioned, EMF obtain their carbon from living trees. Glucose and other monosaccharides are transferred from trees to the EMFs through the ectomycorrhizae. By this way, more than half of the photosynthates of a seedling and up to 21% of the photosynthates of a mature tree are allocated to their symbiotic partner. Trees growing together share almost 40% of the total carbon [55]. The diversity of tree species concerning their water and carbon fluxes in a mixed forest ecosystem is reflected in the carbon isotope composition of the photosynthetic assimilated organic matter [56]. The *δ*^13^C of organic matter is influenced by many environmental factors, including light intensity, atmospheric CO_2_ levels and water availability [57]. Furthermore, mycorrhizal fungi are enriched in ^13^C compared to their host trees, which is consistent with fungi receiving up to 20% of the total carbon fixed by their host trees [58,59]. In summary, forest ecosystems are driven by their complex settings [60], thus making it not possible to discriminate geographical origin among truffles based on their carbon isotope signatures. The *δ*^15^N and *δ*^13^C values in Slovenian samples are comparable to other samples and ranged from 4.2‰ to 19.6‰ and from −28.5‰ to −24.5‰, respectively. The highest *δ*^15^N and the lowest *δ*^13^C values were also observed in *T. magnatum* from Slovenian Istria, while the highest *δ*^13^C value was observed in Dinaric region (Marija Snežna). The *δ*^13^C value of −26.6‰ observed in Snežna jama is indeed enriched in ^13^C compared to the plant material available at this location ranging from −32.8‰ to −27.0‰ [61].

A broad range of *δ*^34^S values (−15.4‰ to +11.3‰) was also observed (Appendix A). The lowest and the highest *δ*^34^S values were recorded in the Italian samples, which is consistent with the high heterogeneity of Italian forest ecosystems with high fungal biodiversity [62]. Typically, 95% of the forest soil sulphur is in organically bound forms such as the ester sulphate, which is synthesised by soil microorganisms, and carbon-bonded S [63,64], where plants take up sulphur primarily as the sulphate anion, SO_4_^2−^ [65]. The incorporation of sulphur into the fungi is, therefore, influenced by root access to SO_4_^2−^ [66]. Since there is little or no fractionation of the sulphur isotopes in plant metabolism, plants will have *δ*^34^S values reflective of those in rainwater, which in turn will influence the isotopic ratio of sulphur in the truffle. However, sulphur not only has multiple biological roles, but it is also a key component of volatile substances that add to the unique truffle aroma such as dimethyl sulphide, which is the predominant aroma compound in black truffle [67]. Recently, two new sulphur compounds were identified in the aroma of black truffle, 1-(methylthio)propane and 1-(methylthio)-1-propene, while a key compound responsible for white truffle aroma is bis(methylthio)methane [14,67]. To date, no indication of the range of *δ*^34^S values in the aroma compounds of truffles has been reported in the literature. Thus, we believe that such a high range in *δ*^34^S values and especially very low *δ*^34^S value of −15.4‰ found in *T. magnatum* from Perugia could be related to sulphur metabolic pathway in truffles that needs to be further investigated.

Delta ^2^H values range from −56.0‰ to 14.8‰ (mean = −15.8‰ ± 13.0‰), and *δ*^18^O values from 15.8‰ to 22.5‰ (mean = 19.4‰ ± 1.3‰). The lowest *δ*^2^H (mean = −47.4‰ ± 9.7‰) and *δ*^18^O (mean = 16.5‰ ± 0.7‰) values were recorded in samples from China. In Slovenian truffles, higher *δ*^2^H and *δ*^18^O values were observed in Sežana due to the mild and dry climate comparing to other regions. Figure 5a reveals how the isotopic ratios of these two elements are tightly linked, providing information about the composition of environmental water. The observed slope is similar to a slope of eight (*δ*^2^H = 8 × *δ*^18^O + 10) in the meteoric water relationship [68], indicating that both isotopes originate from local meteoric water and that metabolism and biosynthesis are of minor importance. Hydrogen and oxygen fixed in the same tissues may be a mixture of atoms derived from body water and water-tissue fractionation during biosynthesis.

Trees can develop deep root systems, the activity of which can be traced using the stable isotopes of water (*δ*^2^H and *δ*^18^O). Sources from which trees take up water (soil water at different depths, fog, dew, and groundwater) tend to have different isotopic compositions due to evaporative fractionation and the rainout effect and as a result of carbon fluxes in mixed forest ecosystems through the assimilation of CO_2_ during photosynthesis [69,70]. It is thought that isotopic fractionation does not occur during water uptake by plants, which means it is possible to identify the source of the water, i.e., rainwater or groundwater. Interestingly, Barbeta et al. [71] recently raised the possibility of fractionation during root-soil interactions, taking into account the effect of different soil and plant root characteristics on the exchange of water, carbon and the atmosphere. The authors observed that *Quercus robur* used deeper soil water with more negative *δ*^2^H and *δ*^18^O values than *Fagus sylvatica*, which typically has a shallower root system. It is interesting to also note that truffle samples associated with *Quercus robur* have lower *δ*^2^H and *δ*^18^O values comparing to truffles associated to *Fagus sylvatica* (Figure 5b). Water access also depends on soil porosity. Thus, *δ*^2^H and *δ*^18^O values of the truffles will record the geographic information associated with a specific location. In theory, the *δ*^2^H and *δ*^18^O signatures in truffles could be affected not only by changes in transpiration but also by the photosynthetic reactions occurring in the host trees [69]. 

Considering these findings, it is likely that isotopic signatures in truffle ascocarps depend on the dominance of tree species in mixed forests. *T. aestivum* is mainly associated with the *Quercus* spp., *Carpinus betulus*, *Betula pendula*, *Fagus sylvatica*, and *Corylus avellana* trees (Appendix A), thereby showing different *δ*^2^H and *δ*^18^O isotopic signatures (Figure 5b). Collectively, a combination of precipitation and temperature affects the functioning of the forest ecosystem, profoundly changing the environment in which truffles grow.

### 2.3. Strontium Stable Isotope Ratios

Isotope application of heavier elements such as Sr can provide additional information on geographical origin since plants inherit the isotopic signature from their geological and pedological environment [72]. In order to improve the geographical discrimination of truffle samples, we analysed ^87^Sr/^86^Sr values and combined them with Sr levels and Rb/Sr, Sr/Ca, Ba/Ca, and Mg/Ca molar ratios (Table 2). 

A wide range of Rb/Sr, Sr/Ca, Ba/Ca, and Mg/Ca ratios of the truffles were observed (Table 2). The highest Sr/Ca and Ba/Ca ratios were determined in *T. magnatum* from Lukini (Slovenian Istria) compared to the rest of samples, regardless of geographical origin and species affiliation. The highest Mg/Ca ratios were also observed in Slovenian samples *T. brumale* from Marija Snežna, followed by *T. magnatum* from Lukini and *T. aestivum* from Rajndol. A high Mg/Ca ratio is related to dolomite weathering, while a low Mg/Ca ratio typically less than 0.1 corresponds to calcite weathering conditions. The bedrock in Slovenia is primarily composed of Mesozoic carbonates (limestone and dolomite) and siliclastic sediments exposed near the surface, especially in areas with high topographic relief. The majority of dolomite bedrock is found in the Dinaric karst region. The siliclastic sediments are primarily Late Paleozoic, the limestone rocks are primarily Triassic, and the dolomites are primarily Jurassic in age. As is seen, most Slovenian locations are located in the Dinaric karst region, where dolomite prevails. 

The proportion of ^87^Sr to total Sr increases at a rate dependent on the available Rb in soil minerals. Accordingly, geological regions rich in Rb relative to Sr will have a high ^87^Sr/^86^Sr ratio, while regions with low Rb relative to Sr will retain low ^87^Sr/^86^Sr ratios for long periods of geological time. It was found that the Rb and Sr concentrations in Slovenian truffles were in the range from 1.16 to 25.7 mg/kg and 1.45 to 18.8 mg/kg, respectively (Table 2, Appendix A). The highest Rb/Sr concentration ratio was observed in samples from Marija Snežna. The ^87^Sr/^86^Sr isotope ratio values in Slovenian truffles ranged from 0.70862 to 0.71375. Most of the samples were in the range from 0.710 to 0.713. The lowest ^87^Sr/^86^Sr values were determined in Sežana and Marija Snežna, 0.70862 and 0.70868, respectively. The highest ^87^Sr/^86^Sr ratio was determined in truffles from Bloke, a karst plateau, composed mainly from limestone and dolomite [73]. It is notable that truffles from Žlebič had lower ^87^Sr/^86^Sr isotopic ratios compared to truffles from Bloke and Rajndol. The central parts of Slovenia are mainly covered in Quaternary terrestrial deposits (gravel and sand), potentially contributing to high ^87^Sr/^86^Sr isotopic ratios of truffles from Spodnje Blato, Pluska, and Meja. Comparing the ^87^Sr/^86^Sr ratios in the analyzed Slovenian truffles with the predictions of the ^87^Sr/^86^Sr ratios in the bioavailable fraction of the soil [74], it is evident that in general there is a certain degree of correlation between truffles and soil. The exceptions were the truffles from Bloke that had much higher isotope ratio than the predicted value of the soil. However, it should be noted that the soil analyzed by Hoogewerff et al. [74] is farmland which is often treated with lime and fertilizer, which can alter strontium composition [75], whereas the truffles presumably are collected in forests that receive no or minimal treatment. Thus the baseline maps made using farmland soil samples may not reflect the strontium isotopic composition of forests in a given area. 

From the foreign truffle samples, the highest ^87^Sr/^86^Sr ratio of 0.71219 was determined in truffles form Cantavieja, Spain. Truffles from Perugia, Italy and Poland had low ^87^Sr/^86^Sr ratios. The geological deposits of Umbri, a where Perugia (500 m above sealevel) is located, consists largely of limestone (formed in the ocean) that has the ^87^Sr/^86^Sr ratio of the ocean at the time of formation [76,77]. In Poland, *T. aestivum* are dominant in the southern part, where the lithology comprises Jurassic and Cretaceous limestone and marlstone on rendzic leptosols [78], having intermeditate Sr isotopic values, 0.706–0.709 [79,80]. The relief of the Šipovo region is mostly formed from the Lower to Middle Devonian sedimentary material composed of lime rocks and dolomite [81]. Truffles *T. indicum* are native to southern China, in the Sichuan and Yunnan provinces where terrain is dominated by metamorphic or igneous rocks [82]. This could explain the lowest Rb/Sr ratio in the Chinese truffles tested.

The large variability of Sr ratios in truffles reflects the wide diversity of the local biogeochemistry of the environment in which truffles grow. This could be useful for the assignment of the geographic origin of truffles. For example, for truffles coming from different regions within the same climate zone (and therefore having similar *δ*^2^H and *δ*^18^O values), supplementary information about the ^87^Sr/^86^Sr isotope ratio may contribute an additional level of geographical resolution, provided that different lithologies exist within a certain region. As the ^87^Sr/^86^Sr ratio plays a significant role in authenticating the geographical origin of environmental and food matrices, it is important to identify where the Sr fingerprint in truffles comes from. Therefore, for a statistical approach to the geographical origin of truffles, a large dataset of precise and accurate ^87^Sr/^86^Sr values is needed in order to evaluate the indicator variability range of both the truffle and the soils and to build classification models. Despite the small sample size of the present study, the determined ^87^Sr/^86^Sr values indicate, as expected, that truffles reflect the local geochemistry of the environment in which they grow. Although it is difficult to determine quantitatively which Sr source exerts dominant control over ^87^Sr/^86^Sr in truffles, the ^87^Sr/^86^Sr of truffles appeared at the first glance to be controlled by the carbonate fraction of soil, which makes Sr most useful for determination proveniance in areas without limestone. It should also be pointed out that this approach is associated with uncertainties and site-specific challenges, since the mycorrhizal relationships with trees and mineral weathering are equally complex, often involving varying combinations of Sr sources and isotopic signatures, thus making determining the truffle’s geographical origin a yet greater challenge.

### 2.4. Geographical Discrimination of Truffle Samples

Multivariate discriminant analysis (DA) was used to classify truffle samples based on elemental compositon and isotopic values. Stable isotopes and elemental fingerprinting were then used to determine the geographical origin of truffles collected from seven countries: Slovenia (n = 31), Italy (n = 6), North Macedonia (n = 5), Croatia (n = 3), Poland (n = 3), China (n = 3), and Spain (n = 2). The statistical method was used to check the two- and three-dimensional charts to test if the groups to which the observations belong are distinct and to show the properties of the groups using explanatory variables. 

A confusion matrix was also constructed to describe the classification performance. This method can be used to create a predictive framework. After optimization of the model, including 33 variables, five variables were excluded: Mg/Ca, Cd, Mn, ^87^Sr/^86^Sr and Rb/Sr. Therefore, the analysis was performed using 28 variables belonging to the origin classes: SLO (Slovenia), IT (Italy), MK (North Macedonia), CRO (Croatia), PL (Poland), CN (China), and ES (Spain). Figure 6 shows a two-dimensional chart with centroids and confidence circles at a significance level of 5%. Cross-validation resulted in 77% correct classification.

The geographical discrimination of truffle samples is also presented in Orange Visual Programming by using linear projection. A different colour represents each origin class, and the score was computed as follows: for each data instance, the method finds the ten nearest neighbours in the projected 2D space, that is, on the combination of attribute pairs. It then checks how many of them have the same colour. The total score of the projection is then the average number of same-coloured neighbours. Computation for continuous colours is similar, except that the coefficient of determination is used to measure the local homogeneity of the projection. The model gives the same discrimination result as obtained by the XLSTAT software, but more importantly, it gives also an excellent graphical projection of the importance of each variable (Figure 7).

Projections of unit vectors, that is, their corresponding anchors, that are very short compared to the others indicate that their associated attribute is not very informative for a particular classification task. The most important anchors (variables) that separate Polish truffles from other countries truffles are Sr, *δ*^13^C, Ba, and V. These parameters could be related to different forest ecosystems and soil properties that exhibit higher concentrations of Sr and low concentrations of Ba and V. The *δ*^13^C values are also higher. Samples from Croatia and North Macedonia are similar, but both groups are well separated from the other countries based on their Pb, Sr, Ni, Cr content and *δ*^18^O values. Chromium and *δ*^18^O are also important for distinguishing between these two groups. In particular, *δ*^18^O could be related to different climatic conditions, since the North Macedonia truffles were collected at altitude (>2000 m). Climatic conditions are also an important factor that separates Slovenian truffles from the Italian, Spanish and Chinese truffles. However, between these groups, separation is possible based on the Ba/Ca ratio and Sr/Ca, indicating different soil properties and geology. The last anchor separates Italy, Spain and China. Although other variables are less important, they still contribute to the overall good discrimination and must be included in the model, while Cd and Mn should be excluded. The reason why these two elements do not contribute to the good discrimination of the groups remains unclear. For example, Cd concentrations are highly variable in the samples and as Cd is volatile, it may be that high Cd samples are from areas with more airborne pollution, which does not really correlate with any specific geographic origin.

### 2.5. Species Discrimination of Truffle Samples

A model was also developed for discriminating truffles based on species. The analysis included 16 different variables belonging to the six origin classes: TUBAES (*T. aestivum*), TUBBRU (*T. brumale*), TUBMAG (*T. magnatum*), TUBMEL (*T. melanosporum*), TUBMES (*T. mesentericum*), and TUBIND (*T. indicum*). Certain species were collected only from one country; therefore, to eliminate of the effect of the geographical origin, several parameters were excluded, since their signature relates to geological and pedological environment (*δ*^34^S, Ba, Ba/Ca, Cs, *δ*^18^O, *δ*^2^H, Sr, Mg/Ca, Rb, *δ*^13^C, and Sr/Ca, ^87^Sr/^86^Sr and Rb/Sr), or pollutants (Pb, Al, Cd, and Hg). Figure 8 shows a two-dimensional chart with centroids and confidence circles at a significance level of 5%. Cross-validation resulted in a 74% correct classification. 

This is only the preliminary study with a limited number of samples present in certain classes. Nevertheless, the discrimination of some specific species such as for *T. magnatum* (100% correct classification) and *T. aestivum* (87% correct classification) gives satisfactory results. The key variables that distinguish *T. magnatum* are *δ*^15^N, V and Zn. The Zn content of *T. magnatum* was twice that of the other truffle species (Figure 3), which supports Popović-Djordjević et al.’s [34] finding that *T. magnatum* can assimilate/accumulate Zn. Unfortunately, in their study, V was not determined. High *δ*^15^N values determined in *T. magnatum* could be related to differences in the source of organic N or the depth of N acquisition. *Tuber brumale* and *T. melanosporum* can be distinguished based on *δ*^15^N values and the levels of Na, Cu, S and P. 

Also, these two species exhibit higher *δ*^15^N values compared to other species, but not as high as *T. magnatum.* These values can be explained by different nutrient sources, although isotopic fractionation during the transport of nitrogenous compounds from truffles to their host plants cannot be excluded. *Tuber brumale* can also assimilate/accumulate Na, Cu, S and P [34]. The assimilation of S is likely related to sulphur metabolism in truffles, which appears to be active in ascocarps of *T. melanosporum,* especially the part related to the production of volatile substances [83]. *Tuber aestivum*, *T. mesentericum* and *T. indicum* are separated from other species based on the levels of Ni, Cr, Mn, Mg, As, and Cu, which are related to regional soil properties. However, to characterise different truffle species more precisely, the analysis of a higher number of countrywide representative samples is needed. In addition, the differentiation according to the species could be further improved if other parameters such as volatile organic compounds, esters, amino acid, organic acids were included using appropriate chemometric tools.

## 3. Materials and Methods

### 3.1. Sample Collection

Samples (n = 58) of truffle species (*Tuber aestivum* (40), *Tuber melanosporum* (5), *Tuber mesentericum* (3), *Tuber brumale* (2), *Tuber indicum* (3), *Tuber macrosporum* (1), and *Tuber magnatum* (4)*)* were collected directly from their natural and cultivated habitats from seven countries—SLO (Slovenia), IT (Italy), MK (North Macedonia), CRO (Croatia), BIH (Bosnia & Herzegovina), PL (Poland), CN (China), and ES (Spain)—during harvest season (from August 2018 to February 2019) (Figure 9). The details of harvesting dates are included in Appendix A. To discriminate between species and origin multi-elemental composition and stable isotope ratio analyses were performed. Further, fifteen samples were selected for determining the natural variation of ^87^Sr/^86^Sr isotope ratios of the truffles.

### 3.2. Reagents, Standards, Calibration Solutions and Samples

Ultrapure water (18.2 MΩ cm) was obtained using a Milli-Q Element System (Merck Millipore, Watertown, MA, USA). Nitric acid (HNO_3_, Suprapur 65%), hydrogen peroxide (H_2_O_2_, Emsure ISO 30%), and hydrofluoric acid (HF, Suprapur 40%) were purchased from Merck (Darmstadt, Germany). All polypropylene materials were cleaned by soaking them in 10% (*v*/*v*) HNO_3_ solution, thoroughly rinsed with MilliQ water and dried before use. 

An inductively coupled plasma (ICP) multi-element standard solution XXI (10 mg/L) containing the following elements: Al, As, Ba, Ca, Cd, Co, Cr, Cs, Cu, Fe, K, Mg, Mn, Na, Ni, Pb, Rb, Sr, V, and Zn, was obtained from Merck (Darmstadt, Germany). A standard solution of Hg (10 mg/L) was prepared separately. A series of ICP single-element standards (1000 mg/L) of P, S, Sr, and Rb, were also obtained from Merck (Darmstadt, Germany). Strontium and Rb standards were used only after a chromatographic extraction procedure. Since no suitable certified and standard reference material is available for fungi, the accuracy of the sample pretreatment method was assessed using the two certified reference materials: tomato leaves (NIST 1573a) and peach leaves (NIST 1547), both acquired from the National Institute of Standards and Technology, NIST (Gaithersburg, MD, USA). 

The accuracy of the *δ*^13^C, *δ*^15^N, and *δ*^34^S determination was checked with the following international and laboratory reference materials: ammonium sulfates IAEA-N-1 (*δ*^15^N = +0.43‰ ± 0.07‰) and IAEA-N-2 (*δ*^15^N = +20.41‰ ± 0.16‰), barium sulfate NBS 127 (+21.12‰ ± 0.22‰), casein OAS (Sercon; *δ*^13^C = −26.98‰ ± 0.13‰, *δ*^15^N = +5.94‰ ± 0.08‰, *δ*^34^S = +6.32‰ ± 0.8‰), and casein IAEA-CRP (*δ*^13^C = −20.3‰ ± 0.09‰, *δ*^15^N = +5.62‰ ± 0.19‰, *δ*^34^S = +4.18‰ ± 0.74‰). The CBS (Caribou Hoof Standard) with *δ*^2^H values of −157‰ ± 0.9‰ and *δ*^18^O values of 3.8‰ ± 0.3‰, and KHS (Kudu Horn Standard) with *δ*^2^H value of −35.3‰ ± 1.1‰ and *δ*^18^O value of 20.3‰ ± 0.3‰ standards were used for δ^2^H, and δ^18^O determination.

For Sr-matrix separation, a Sr-specific resin (TrisKem International, Bruz, France) was used. For ^87^Sr/^86^Sr isotope ratio analysis, all samples were analysed in a sample-standard bracketing sequence with a Sr isotopic standard solution of NIST SRM 987 SrCO_3_ (strontium carbonate, ^87^Sr/^86^Sr = 0.71034 ± 0.00026, NIST, Gaithersburg, MD, USA).

### 3.3. Sample Pretreatment Method

Soil residues were removed from the truffles either by mechanical cleaning of the fruiting body or by washing with Milli-Q water. After the cleaning, the fruiting bodies were cut into 1–2 mm thick slices using a ceramic knife and then freeze-dried. The analysis was performed using the peridium. After lyophilisation, the samples were ground and homogenised before analysis. 

Each freeze-dried sample (0.10 g) was weighed directly into a teflon microwave digestion vessel, to which was added 2 mL of HNO_3_. The sample was then digested using an UltraWAVE™ microwave system (Single Reaction Chamber Microwave Digestion System, Milestone, Soristone, Italy). The program was as follows: a 20-minute temperature increase to 240 °C, held for 15 min at 100 bar and then allowed to cool to room temp. Two replicates were prepared for each sample. Certified reference materials and blank samples were also prepared. 

Each solution was quantitatively transferred into a 10 mL polyethylene graduated vials and filled up to the mark with Milli-Q water. In order to remove any residuals, samples were filtered through Millex-HV syringe filters (0.45 µm, Millipore hydrophilic PVDF filter membrane; Merck Millipore Ltd., Tulagreen, Ireland) and stored at 4 °C until analyses. After each mineralisation cycle, a cleaning cycle was performed with 2 mL of HNO_3_:H_2_O (1:1, *v*/*v*) to eliminate cross-contamination. 

### 3.4. Analytical Procedure and Instrumentation

Multi-elemental (ICP-MS) analysis and stable isotope determination (*δ*^13^C, *δ*^15^N, *δ*^34^S, ^87^Sr/^86^Sr) was performed at the Department of Environmental Sciences, Jožef Stefan Institute in Ljubljana, while the determination of *δ*^18^O and *δ*^2^H values was obtained at the Department of Food Quality and Nutrition, Research and Innovation Centre, Fondazione Edmund Mach in San Michele all’Adige, Italy. 

#### 3.4.1. Inductively Coupled Plasma-Mass Spectrometry (ICP-MS)

Measurements were performed on an Agilent 8800 triple quard instrument (ICP-QQQ, Agilent Technologies, California, USA). Twenty-three elements (Na, Mg, Al, P, S, K, Ca, V, Cr, Mn, Fe, Co, Ni, Cu, Zn, As, Rb, Sr, Cd, Cs, Ba, Hg, and Pb) were determined using a multi-element method. Multi-element standards for external calibration was prepared using the ICP standard solution XXI and single-element standards (P and S), which were diluted with MQ water with the addition of HNO_3_ (5%, *v*/*v*). Two external calibration curves were prepared. A six-point calibration covered the range between 0 and 10 ng/g for Hg, while nine-point calibration curve covered the range between 0 and 250 ng/g for other elements investigated.

All measurements were made under strict quality control procedures. Blank samples and reference materials were run together with the samples daily. The limits of detection (LOD) were calculated as three times the standard deviation of blank noise. The mean values of the replicate sample measurement were used for data analysis. Levels of Al, Ni, and Pb in some samples were below LOD. However, elements were not excluded if a significant number of samples were concentrated markedly above the LOD level as they may be characteristic of provenance. The LODs for Na, Mg, Al, P, S, K, Ca, V, Cr, Mn, Fe, Co, Ni, Cr, Zn, As, Rb, Sr, Ni, Cd, Cs, Ba, Hg, and Pb was 5.5, 1, 3.5, 5.5, 5, 20, 6, 0.013, 0.05, 0.02, 0.7, 0.005, 0.1, 0.2, 0.75, 0.007, 0.01, 0.08, 0.0025, 0.003, 0.06, 0.004, and 0.025 mg/kg of truffle sample, respectively.

#### 3.4.2. Quantitative-XRF Analysis

Micro-XRF imaging of freeze-dried tuber hand cross-sections (1 × 1 × 0.1 cm) was performed at the Synchrotron Light Research Institute, Thailand at the BL6b beamline. The BL6b utilized continuous synchrotron radiation from the bending magnet with energy range from 2–8 keV and the beam size of 100 μm at the sample position using polycapillary half-lens as the X-ray optics. To perform the measurements, the sample was placed on a three-degrees of freedom high precision motorized stage in air. The XRF signal was collected by an AMPTEK single-element Si (PIN) solid-state detector with thin Be window and energy resolution of 150 eV at 5.9 keV [84]. 

The tuber cross-sections were mounted between two layers of a 4 µm Mylar foil stretched on an Al frame and raster scanned with a 100 µm polychromatic beam and a 100 µm step-size. The X-ray fluorescence spectra obtained in each pixel were batch fitted by PyMCA [85]. Due to the complexity of polychromatic synchrotron excitation, quantification was performed according to Zidar et al., (2016) [86]. In short, the average intensity was calculated from the intensities of specific X-ray emission lines (Ca-Ka, Fe-Ka, Mn-Ka, Cu-Ka and Zn-Ka) obtained after fitting of XRF spectra by PYMCA in each pixel and assigned to the total Ca, Fe, Mn, Cu or Zn concertation measured in particular cross-section by energy dispersive X-ray fluorescence spectrometry (EDXRF) [87]. Prior to the EDXRF measurement, each particular cross-section was homogenised and pressed to a pellet with a diameter of one centimetre. The EDXRF was performed using an energy dispersive X-ray spectrometer, equipped with a Cd-109 radioisotope source and a Si(Li) detector (Canberra, 157 Meriden, USA). The XRF measurements were performed in air. The analysis of the XRF spectra was performed according to Nečemer et al. [87], and quantification according to Kump et al. [88]. Quality assurance for the element analyses was performed using standard reference materials: NIST SRM 1573a (tomato leaves as a homogenised powder) and CRM 129 (hay powder) were both analysed in the form of pressed pellets.

### 3.5. Stable Isotope Ratio Analysis of Light Elements

The stable isotope ratios measurements were performed using isotope ratio mass spectrometry (IRMS) and expressed in the *δ*-notation in ‰ according to Equation (1) [89]:(1)δ(i/jE)=δi/jE=i/jRP−i/jRRefi/jRRef
where superscripts *i* and *j* denotes the higher and the lower atomic mass number of element *E*, *R_p_* and *R_Ref_* represent the ratios between the havier and the lighter isotope (^2^H/^1^H, ^13^C/^12^C, ^15^N/^14^N, ^18^O/^16^O, ^34^S/^32^S) in the sample (*P*) and reference material (*Ref*), respectively. The *δ*^2^H and *δ*^18^O were reported relative to the V-SMOW (Vienna-Standard Mean Ocean Water) standard, *δ*^13^C values were reported relative to the V-PDB (Vienna-Pee Dee Belemnite) standard, while *δ*^15^N and *δ*^34^S were reported relative to AIR and the V-CDT (for Vienna Cañon Diablo Troilite) standard, respectively [89]. 

The *δ*^13^C, *δ*^15^N and *δ*^34^S values in freeze-dried samples were determined simultaneously using an IsoPrime–100 Vario PYRO Cube (OH/CNS) Pyrolizer/Elemental Analyzer (IsoPrime, Cheadle Hulme, UK). Approximately 4 mg of the sample and 4 mg of tungsten oxide (WO_3_) were weighted into a tin capsule, sealed and placed into the automatic sampler of the elemental analyser. Each sample was measured in triplicate, and the average values was considered. The results were normalized against the following international and laboratory reference materials: IAEA-N-1 and IAEA-N-2 for nitrogen; IAEA-CRP-2013 and Casein OAS B2155 Sercon for carbon and suphfur, respectively.

The ^2^H/^1^H and ^18^O/^16^O values were determined by transferring 0.2 mg of dry truffles into a silver capsule and analysing the sample simultaneously using TC/EA pyrolyser (Thermo Finnigan) coupled to a DELTA XP isotope ratio-mass spectrometer, IRMS (Thermo Scientific). For normalisation of the results, two internal laboratory reference materials were applied: CBS (Caribou Hoof Standard) and KHS (Kudu Horn Standard). Measurements precision was 0.2‰ for *δ*^13^C and *δ*^15^N, 0.3‰ for *δ*^34^S and *δ*^18^O and 1‰ for *δ*^2^H.

### 3.6. Stable Isotope Ratio Analysis of Heavy Elements

Sr isotope ratio determinations were performed using the Nu II multicollector ICP-MS instrument (Nu Instruments, Ametek Inc., United Kingdom). After the microwave digestion, samples were preconcentrated by evaporation to near dryness. The residuals were dissolved in 1 mL of 8 M HNO_3_ and the Sr separated from the matrix using a Sr-specific resin. For this, a column was filled with 0.30 g of the resin, which was activated by performing several washing and elution cycles (Appendix A).

### 3.7. Statistical Analysis

Statistical analysis included one-way ANOVA and Duncan’s test. Probability (p) values of less than 0.05 were used to indicate a significance level. If a significance was noted in a response factor, the calculation was followed by post-hoc testing using the Tukey’s Honestly Significant Difference (HSD) test. For non-normally distributed data, a one-way analysis of variance by ranks (Kruskal–Wallis test) was performed.

Further, to identify those parameters that can discriminate truffles according to the geographical origin and/or variety, a multivariate discriminant analysis (DA) was used. The data were evaluated using the statistical software packages XLSTAT (Addinsoft, NY, USA), Orange Visual Programming (University of Ljubljana, Ljubljana, Slovenia) and OriginPro 2018 (OriginLab Corporation, Northampton, MA, USA). 

## 4. Conclusions

A wide variability of the element concentrations within and among truffle species and locations/countries was observed. Despite the fact that the Slovenian truffles shared some similar characteristics with the samples originating from other countries, differences in the element concentrations suggest that the respective truffle species may respond selectively to nutrients from a certain soil type under environmental and soil conditions. The heterogeneity of geographical and/or environmental factors influencing the element composition of truffles, therefore, enable the possibility to discriminate between geographical origin/species. It was found that a combination of elemental and isotopic profiling coupled to multivariate analysis is a promising tool for characterizing truffles according to geographical origin and species. However, the classification model performance must be improved by increasing the size of the dataset and include other natural tracers such as strontium isotopes ratios, since ^87^Sr/^86^Sr ratios differ according to the natural geology of the region of production and makes Sr most useful for determination proveniance in areas with and without limestone. Future research should also investigate the maturation stage, water source and availability, in association with specific host tree species that can influence elemental and stable isotope composition of truffles. It is expected that the proposed approach will aid in protecting consumers and truffle producers from fraudulent labeling regarding species affiliation and geographical origin. 

## Figures and Tables

**Figure 1 molecules-25-02217-f001:**
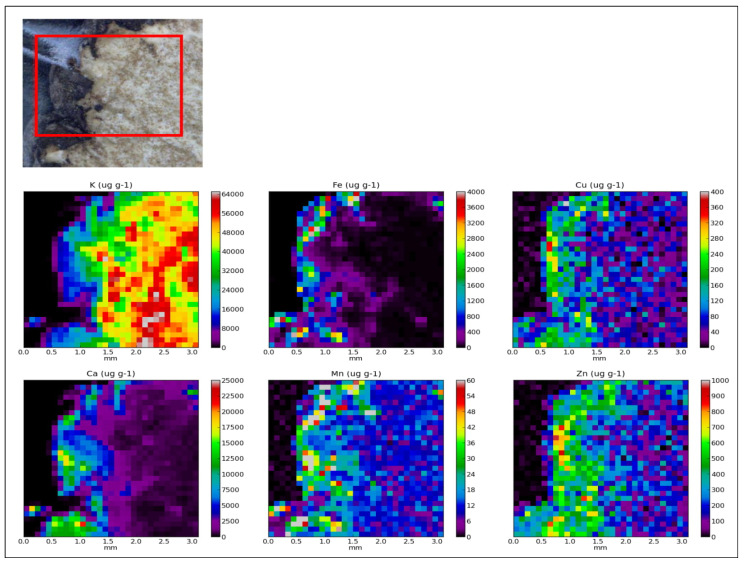
XRF quantitative analysis and Micro-XRF analysis of *T. aestivum* recorded using the BL6b, SLRI, Polychromatic Beam (Synchrotron Light Research Institute) with 100 µm lateral resolution.

**Figure 2 molecules-25-02217-f002:**
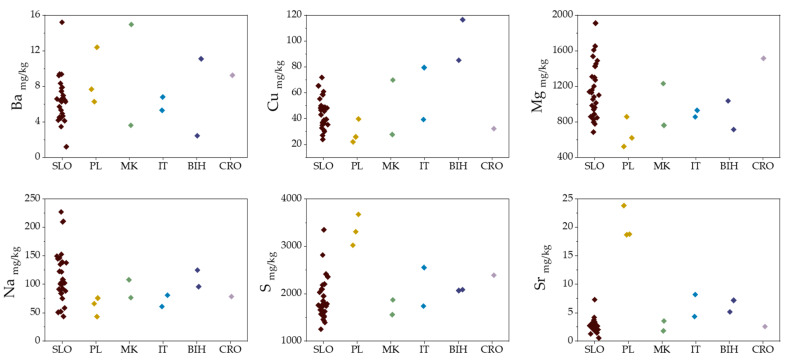
Column scatter plots of element contents in *T. aestivum* from different geographical regions: SLO (Slovenia), PL (Poland), MK (North Macedonia), IT (Italy), BIH (Bosnia and Herzegovina), CRO (Croatia).

**Figure 3 molecules-25-02217-f003:**
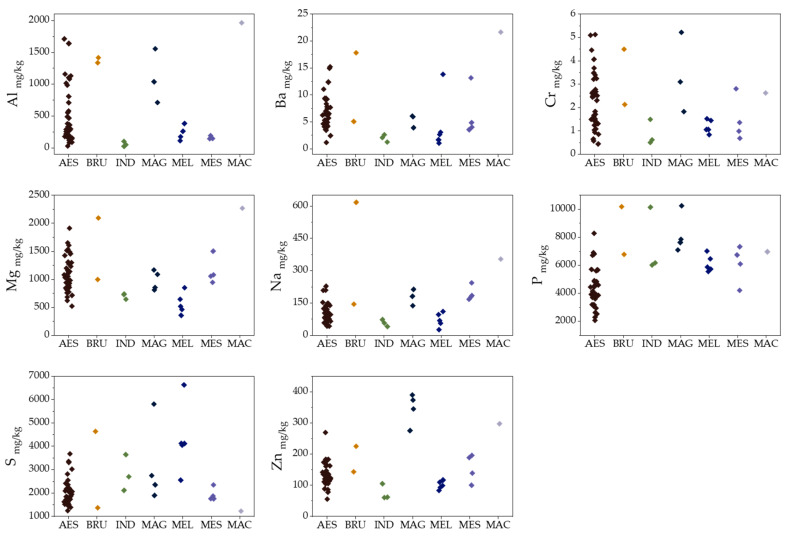
Column scatter plots of element contents in different truffle species: AES (*T. aestivum*), BRU (*T. brumale*), IND (*T. indicum*), MAG (*T. magnatum*), MEL (*T. melanosporum*), MES (*T. mesentericum*), and MAC (*T. macrosporum*).

**Figure 4 molecules-25-02217-f004:**
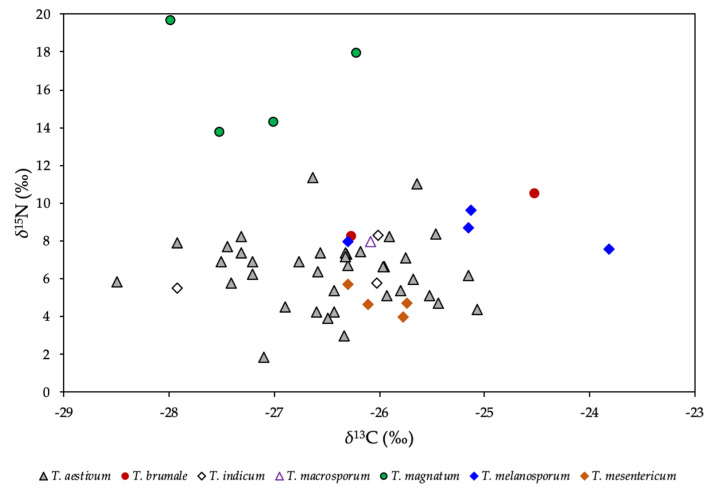
Relationship between *δ*^15^N and *δ*^13^C values in *Tuber* species.

**Figure 5 molecules-25-02217-f005:**
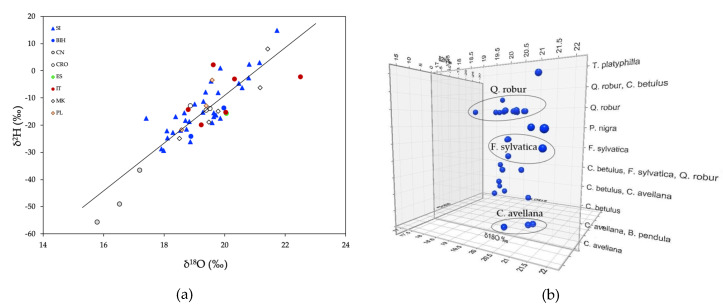
**(a**) A plot of the relationship between *δ*^2^H and *δ*^18^O of truffle samples from eight countries: SLO (Slovenia), BIH (Bosnia and Herzegovina), CN (China), CRO (Croatia), ES (Spain), IT (Italy), MK (North Macedonia), PL (Poland). A line drawn through the data in the plot shows that the data are strongly linked together (y = 8.1x − 171.8; *r*^2^ = 0.75, *p* < 0.001); (**b**) comparison of natural isotopic abundance (*δ*^2^H and *δ*^18^O) of *T. aestivum* in association with host tree species in mixed forest systems (Appendix A).

**Figure 6 molecules-25-02217-f006:**
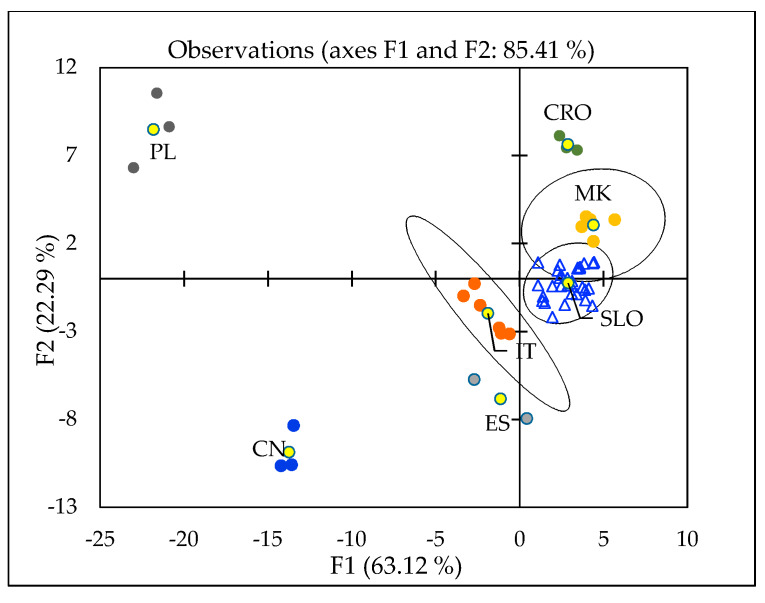
Projection of discriminant analysis (DA) of truffle samples regarding their geographical origin: CN (China), CRO (Croatia), ES (Spain), IT (Italy), MK (North Macedonia), PL (Poland), SLO (Slovenia). Yellow markers refer to centroids.

**Figure 7 molecules-25-02217-f007:**
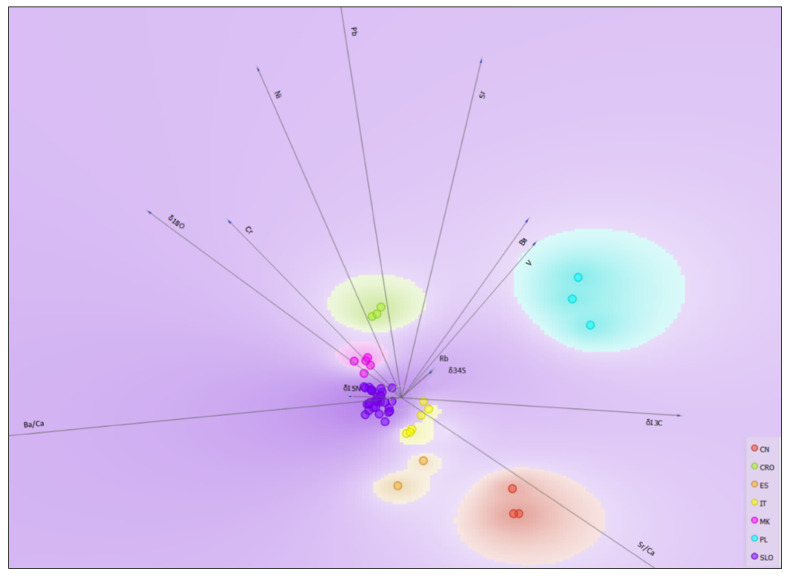
Projection of unit vectors for discriminant analysis (DA) of truffle samples regarding their geographical origin: CN (China), CRO (Croatia), ES (Spain), IT (Italy), MK (North Macedonia), PL (Poland), SLO (Slovenia).

**Figure 8 molecules-25-02217-f008:**
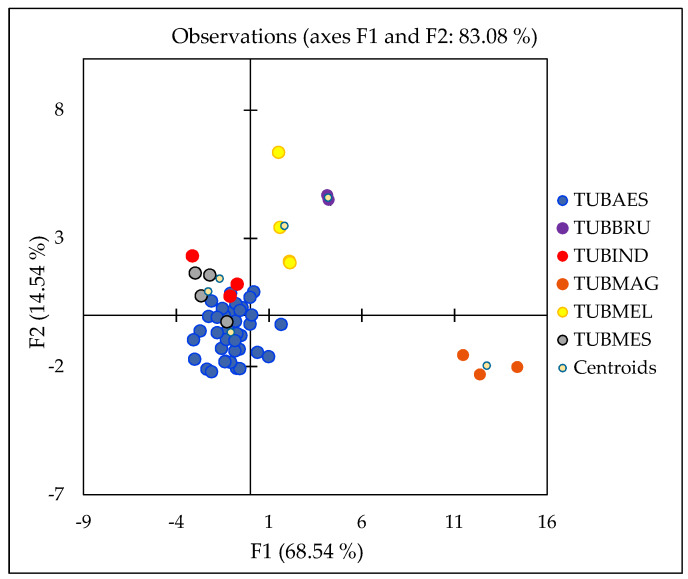
Projection of discriminant analysis (DA) of different truffle species: TUBAES (*T. aestivum*), TUBBRU (*T. brumale*), TUBIND (*T. indicum*), TUBMAG (*T. magnatum*), TUBMEL (*T. melanosporum*), and TUBMES (*T. mesentericum*).

**Figure 9 molecules-25-02217-f009:**
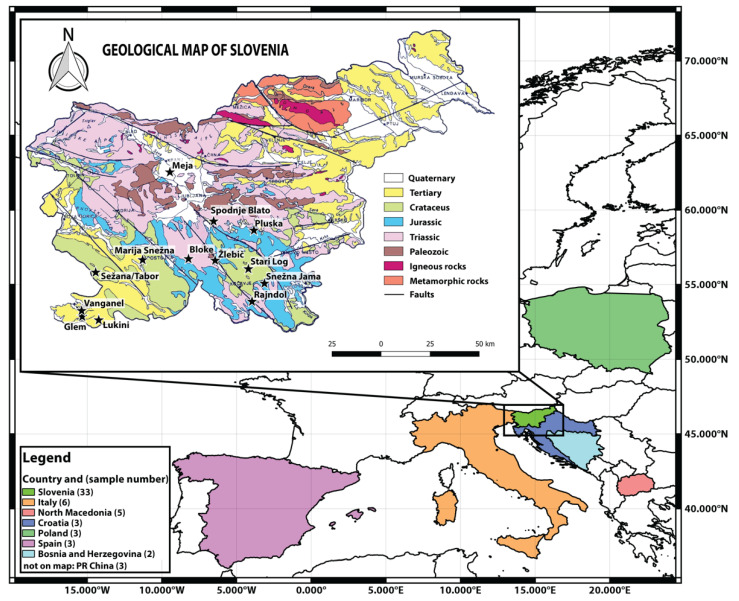
Geologic map of Slovenia (GeoSZ, 2013) with sample sites from the dataset (Appendix A) marked as black stars. In the background are colored European countries with marked natural sites where truffles were collected: Italy, North Macedonia, Croatia, Poland, Spain, and Bosnia and Herzegovina. Only samples from China were purchased on the market.

**Table 1 molecules-25-02217-t001:** Minimum (Min), maximum (Max), and mean concentration (Mean) ± standard deviation (SD) of elements in the peridial layer of the fruiting bodies of different truffle species. All elements are expressed in mg/kg.

Element	*Tuber magnatum* (*n* = 4)			*Tuber melanosporum* (*n* = 5)			*Tuber mesentericum* (*n* = 4)			*Tuber aestivum* (*n* = 39)			*Tuber brumale* (*n* = 2)			*Tuber indicum* (*n* = 3)			*Tuber macrosporum (n = 1)*
	Mean ± SD	Min	Max	Mean ± SD	Min	Max	Mean ± SD	Min	Max	Mean ± SD	Min	Max	Mean ± SD	Min	Max	Mean ± SD	Min	Max	
Al	1103 ± 424	715	1557	233 ± 116	113	382	511 ± 699	143	1559	498 ± 435	29.0	1711	1376 ± 59	1335	1418	59.6 ± 41.1	22.4	104	1961
As	0.20 ± 0.05	0.16	0.26	0.07 ± 0.05	0.03	0.14	0.19 ± 0.20	0.04	0.49	0.20 ± 0.17	0.03	0.72	0.28 ± 0.06	0.23	0.32	0.05 ± 0.01	0.04	0.06	0.28
Ba	8.21 ± 5.87	3.91	16.90	2.13 ± 0.91	1.08	3.08	6.41 ± 4.55	3.56	13.2	6.84 ± 2.96	1.20	15.2	11.5 ± 9.0	5.09	17.8	1.99 ± 0.69	1.26	2.63	21.7
Ca	1594 ± 642	1220	2551	3489 ± 1120	2527	5418	2217 ± 1068	1380	3681	2831 ± 820	668	5135	2653 ± 2365	980	4325	987 ± 274	747	1285	3868
Cd	2.17 ± 1.00	0.77	2.97	2.35 ± 1.75	0.49	4.03	5.49 ± 3.01	1.36	8.50	6.25 ± 3.35	1.78	15.4	9.36 ± 4.46	6.20	12.5	1.30 ± 1.08	0.42	2.51	7.90
Co	0.40 ± 0.13	0.32	0.54	0.17 ± 0.14	0.04	0.37	0.17 ± 0.17	0.07	0.41	0.17 ± 0.12	0.03	0.48	0.32 ± 0.05	0.29	0.36	0.10 ± 0.04	0.07	0.15	0.36
Cr	3.38 ± 1.72	1.82	5.22	1.18 ± 0.29	0.84	1.52	1.46 ± 0.93	0.69	2.80	2.24 ± 1.21	0.44	5.13	3.31 ± 1.68	2.13	4.50	0.87 ± 0.54	0.51	1.50	2.62
Cs	0.12 ± 0.04	0.08	0.16	0.04 ± 0.01	0.02	0.05	0.06 ± 0.07	0.02	0.17	0.07 ± 0.06	0.01	0.23	0.14 ± 0.00	0.13	0.14	0.01 ± 0.00	0.01	0.02	0.19
Cu	61.9 ± 29.1	25.5	94.2	122 ± 119	32.6	270	51.9 ± 8.7	43.3	63.5	46.6 ± 19.0	22.0	116	88.5 ± 80.1	31.9	145	27.5 ± 16.3	17.1	46.4	29.8
Fe	322 ± 84	228	387	156 ± 67	84.2	222	367 ± 492	107	1104	325 ± 284	27.0	1215	646 ± 261	462	831	50.7 ± 30.7	24.5	84.4	589
Hg	0.04 ± 0.00	0.04	0.05	0.05 ± 0.02	0.03	0.08	0.09 ± 0.04	0.06	0.14	0.07 ± 0.04	0.01	0.20	0.03 ± 0.02	0.02	0.05	0.06 ± 0.01	0.05	0.07	0.03
K	31,612 ± 3926	28,203	35,844	20,289 ± 2086	18,194	23,097	21,689 ± 3772	17,194	25,824	21,364 ± 3510	15,464	32,571	26,207 ± 7426	20,956	31,458	23,676 ± 6690	17,051	30,429	23,702
Mg	980 ± 173	813	1167	567 ± 188	360	849	1147 ± 244	947	1503	1090 ± 319	521	1911	1544 ± 777	994	2093	706 ± 52	646	740	2268
Mn	20.0 ± 4.9	15.1	24.6	10.1 ± 4.2	5.36	15.5	20.2 ± 15.3	9.94	43.0	18.2 ± 13.9	5.42	87.3	24.5 ± 12.0	16.0	33.0	8.81 ± 2.80	6.80	12.0	20.8
Na	178 ± 38	138	213	72.0 ± 33.2	27.0	111.0	194 ± 34	168	245	106 ± 44	43.1	227	381 ± 334	145	617	57.7 ± 16.2	41.1	73.5	354
Ni	2.68 ± 1.29	1.36	3.93	0.39 ± 0.18	0.15	0.58	0.79 ± 0.47	0.39	1.46	0.93 ± 0.81	0.23	3.91	1.55 ± 0.28	1.35	1.76	0.36 ± 0.13	0.27	0.51	2.18
P	8209 ± 1400	7091	10252	6124 ± 596	5561	7003	6095 ± 1353	4210	7330	4319 ± 1342	2067	8285	8485 ± 2398	6790	####	7453 ± 2336	6027	10149	6965
Pb	0.59 ± 0.36	0.25	1.08	0.18 ± 0.09	0.08	0.27	0.72 ± 0.67	0.22	1.69	0.48 ± 0.43	0.07	2.33	0.67 ± 0.04	0.65	0.70	0.15 ± 0.16	0.04	0.33	1.01
Rb	10.0 ± 4.4	5.48	14.5	4.95 ± 3.59	1.81	9.18	11.0 ± 5.4	4.88	15.6	9.00 ± 6.38	1.24	28.2	17.8 ± 11.1	10.0	25.7	1.51 ± 0.30	1.16	1.70	16.7
S	3202 ± 1772	1897	5808	4297 ± 1467	2552	6630	1933 ± 285	1752	2353	2014 ± 572	1251	3677	3007 ± 2307	1376	4638	2821 ± 775	2114	3649	1230
Sr	6.12 ± 3.17	3.45	10.7	3.52 ± 2.55	0.81	7.34	2.91 ± 1.43	1.59	4.66	4.43 ± 4.98	0.58	23.8	9.41 ± 9.11	2.98	15.9	3.70 ± 1.25	2.42	4.92	12.0
V	2.28 ± 0.87	1.48	3.21	0.50 ± 0.24	0.25	0.80	0.92 ± 1.18	0.23	2.69	1.20 ± 1.15	0.07	4.95	2.59 ± 0.61	2.16	3.02	0.17 ± 0.09	0.11	0.27	3.37
Zn	346 ± 51	275	390	100 ± 13	83.1	117	156 ± 45	100	196	136 ± 37	55.0	269	184 ± 58	143	225	75.8 ± 25.4	60.7	105	298

**Table 2 molecules-25-02217-t002:** Summary of the ^87^Sr/^86^Sr rato data for truffle species of different geographical origin, associated with local soil geology (Appendix A), concentrations of Sr and molar ratios of Rb/Sr, Sr/Ca, Ba/Ca and Mg/Ca.

Species	Country	Location	Sr _mg/kg_	Rb/Sr	Sr/Ca	Ba/Ca	Mg/Ca	^87^Sr/^86^Sr
TUBAES	SLO	Meja	3.13	0.0019	0.0006	0.0009	0.50	0.71212
	SLO	Pluska	2.28	0.0020	0.0005	0.0006	0.94	0.71088
	SLO	Spodnje Blato	2.66	0.0049	0.0004	0.0006	0.78	0.71094
	SLO	Žlebič	1.45	0.0039	0.0005	0.0008	0.95	0.70985
	SLO	Bloke	2.67	0.0052	0.0005	0.0009	1.05	0.71375
	SLO	Sežana	7.32	0.0019	0.0007	0.0009	0.87	0.70862
	SLO	Rajndol	1.75	0.0076	0.0004	0.0010	1.19	0.71151
	PL	n.d.	18.8	0.0003	0.0025	0.0011	0.41	0.70896
	IT	Perugia	4.35	0.0020	0.0006	0.0005	0.41	0.70894
	CRO	n.d.	2.54	0.0067	0.0005	0.0012	1.13	0.71102
	BIH	Šipovo	7.19	0.0010	0.0011	0.0011	0.56	0.70975
TUBBRU	SLO	Marija Snežna	2.98	0.0088	0.0014	0.0015	1.67	0.70868
TUBIND	CN	n.d.	4.92	0.0002	0.0018	0.0006	0.94	0.70953
TUBMAG	SLO	Lukini	10.7	0.0014	0.0040	0.0040	1.55	0.71105
TUBMEL	ES	Cantavieja	7.34	0.0013	0.0006	0.0007	0.26	0.71219

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
