# Peer review of "Can We Discover Truffle’s True Identity?"

_molecules, 2020, doi:10.3390/molecules25092217_

Round 1

Reviewer 1 Report

The authors used elemental and stable isotope composition analysis to differentiate Slovenian truffles by variety and geographical origin in order to provide tools to counter adulterations. The paper is well written, scientifically sound and i recommend publication as is. 

Author Response

see attachement.

Reviewer 2 Report

The paper presents an interesting experiment of trying to classify truffles by species and geographic location using geochemistry.

This is of interest not only to people working with tracking fraudulent food items but also to people working with trace element and isotope cycling of biological processes.

I recommend that the paper is published after fixing some minor issues.

Below is a list of specific comments: 

Line 19 and other places in the manuscript mainly section 2.3: The strontium analysis is not a ‘stable isotope analysis’. Stable isotopes variation refer to mass-dependent variations between isotopes of a given element. Strontium-87 is produced by the decay of Rubidium-87 and therefor its abundance varies with Rb/Sr of the sample. Variations in strontium-87 are mass-independent, normally this type is called ‘radiogenic isotope analysis’ as strontium-87 is the stable daughter nuclei created by the radioactive decay of rubidium-87.

Stable strontium isotopes, normally refers to the analysis of variations in strontium-84. This should be changed to avoid confusion for the readers.

Line 75: This should be 87Sr/86Sr.

Figure 2+3: It does not make sense to make a box plot for sample populations of 2, 3, 4 and 6 as is the case for most of the countries and species studied here. These boxes should be replaced with individual data-points.

Line 348: One additional complication is that the soil analyzed by ref.75 is farmland which is often treated with lime and fertilizer which can alter strontium composition (see Thomsen & Andreasen, 2019 Science Advances,  DOI: 10.1126/sciadv.aav8083), whereas the truffles presumably are collected in forests that receive no or minimal treatment. Thus the baseline maps made using farmland soil samples may or may not reflect the strontium isotopic composition of forest in a given area.  

Line 353: Perugia is located nearly 500 meter above sealevel. The geological deposits there are not equilibrated with seawater. But the geological deposits of Umbria consists largely of limestone (formed in the ocean) that has the 87Sr/86Sr ratio of the ocean at the time of formation (which is not very different from today).

Figure 6: As strontium-87 is being produced slowly be the decay of rubidium-87 over geologic time, many millions of years, there is no reason to suspect a correlation between strontium isotope ratio and Rb/Sr ratio of the truffles. This figure is not needed.

Line 381: Yes, whenever there is limestone in the soil, anything grown in that area will likely have a limestone signal because of the high Sr content in the limestone and the high degree of solubility of the limestone to other minerals in the soil. This makes Sr most useful for provenancing in areas without limestone.

Line 431: The Cd concentrations are highly variable in the samples and as Cd is somewhat volatile it may be that high Cd samples are from areas with more airborne pollution, which probably does not correlate with any specific geographic origin.

Section 4: It would be helpful here if you added where (in which laboratories) the different analyses were done.

Author Response

See attachement.

Reviewer 3 Report

The manuscript “Can we discover truffle’s true identity?” provides a valuable description of the elemental and isotopic fingerprint of Slovenian truffles. The authors studied the possibility to develop a new model to be used in establishing truffe’s geographical and botanical authenticity. This approach presents a high degree of difficulty by the fact that numerous variables are involved in an attempt to solve a current economic issue.

The authors have reported for δ34S a wide domain, ranging from -15.4‰ to 11.3‰. The -15.4‰ value obtained for a specific Italian truffle is an unusual value indicating a possible error. Since it was no sample preparation a fractionation in this step of analysis could not occurred, leading to the fact that could be a typing error, a chemosynthetic oxidation of the sulphide in the presence of oxygen or some sulfide minerals residue since the peridium was the analyzed part. Could the authors provide an explanation for this odd value or insert a reference to confirm the found value.

The authors should mention the variables that have been used in realizing the discriminant analysis for the geographical origin since the total number is 33 and in the text is specified that only 30 were used for the optimization and then 3 excluded.

From my point of view the section regarding the species discrimination is not correct since the used variables are in general strongly related to the geographical characteristics of the original truffle harvesting area. Also, in order to highlight species differences, other variables such as organic compounds (esters, amino acids, organic acids, etc.) should be added and using mainly samples originating from a single geographical region (e.g. Slovenia only). The present study should mention that some preliminary results are provided regarding the species discrimination and could use only the common varieties from the studied countries.

Simultaneously, a more detailed description of the sample collection is needed since there is a possibility that the harvest season may vary among species and countries, consequently the obtain results for the botanical origin separation could be completely wrong.

Regarding figure 9, I recommend to the authors to eliminate the confidence circle since their shape or distribution is not entirely contained in the image and could cause confusion and to use the same symbol with different colors for the used observations and increase the marker size.

Line 397 and 442 - figures 7 and 9 show a two-dimensional chart since only F1 and F2 are projected.

In the conclusion section as well as in the abstract is mentioned that for a better interpretation of the data other natural tracers, such as strontium stable isotope should be included although Section 2.3 of the present paper represents a section dedicated to the strontium stable isotope ratio data interpretation. Why was this parameter not included in the interpretation of the data in terms of separation according to geographical or botanical origin?

Author Response

See attachement.

Round 2

Reviewer 3 Report

The authors have responded my concerns in a very persuasive manner and introduced new elements to support their conclusions, consequently the present paper has improved considerably. In my opinion, the manuscript entitled " Can we discover truffle’s true identity?", in its current form is very interesting, scientifically sound and should be accepted as it is.